# Hyperleukocytosis in Childhood Acute Leukemia: Early Complications and Survival Outcomes

**DOI:** 10.3390/cancers15123072

**Published:** 2023-06-06

**Authors:** Sirinthip Kittivisuit, Nichanan Jongthitinon, Pornpun Sripornsawan, Natsaruth Songthawee, Shevachut Chavananon, Chompoonut Limratchapong, Edward B. McNeil, Thirachit Chotsampancharoen

**Affiliations:** 1Department of Pediatrics, Faculty of Medicine, Prince of Songkla University, Hat Yai 90110, Thailandchompoonut.lim@gmail.com (C.L.); 2Epidemiology Unit, Faculty of Medicine, Prince of Songkla University, Hat Yai 90110, Thailand

**Keywords:** childhood acute leukemia, early complication, hyperleukocytosis, risk factor, survival outcome

## Abstract

**Simple Summary:**

Hyperleukocytosis (WBC > 100 × 10^9^/L) has been associated with unfavorable outcomes. Few studies have focused on childhood acute leukemia with hyperleukocytosis in developing countries, where the treatment outcomes remain poorer than in developed countries. Our study reviewed the medical records of 690 children who were diagnosed with acute leukemia between January 1998 and December 2017. The incidence of hyperleukocytosis was 16.6% in acute lymphoblastic leukemia (ALL) patients and 20.3% in acute myeloid leukemia (AML) patients. Hyperleukocytosis, extreme hyperleukocytosis (WBC > 200 × 10^9^/L), age less than 1 year, age greater than 10 years, and male sex were independently associated with overall survival in the ALL group, while extreme hyperleukocytosis and age less than 1 year were independently associated with overall survival in the AML group. We believe that to improve survival outcomes for children with hyperleukocytosis, the recommended treatment regimen must be modified, while early treatment-related complications, which are more likely to develop, should raise concerns.

**Abstract:**

Hyperleukocytosis and extreme hyperleukocytosis, defined as initial white blood cell counts greater than 100 × 10^9^/L and 200 × 10^9^/L, respectively, have been associated with unfavorable outcomes. This study aimed to determine the early complications and survival outcomes of childhood leukemia patients with hyperleukocytosis. The medical records of 690 children newly diagnosed with acute leukemia between January 1998 and December 2017 were retrospectively reviewed. The Kaplan–Meier method and log-rank test were used to assess and compare the survival outcomes. Multivariate Cox proportional hazards regression was used to determine associated risk factors for overall survival. We found that 16.6% of 483 childhood acute lymphoblastic leukemia (ALL) patients and 20.3% of 207 childhood acute myeloid leukemia (AML) patients had hyperleukocytosis at diagnosis. ALL patients with hyperleukocytosis had more early complications than those without hyperleukocytosis (*p* < 0.05). Among the ALL group, the 5-year overall survival rate for those with hyperleukocytosis was significantly lower than for those without hyperleukocytosis (37.2% vs. 67.8%, *p* < 0.0001), while the difference was not statistically significant in the AML group (19.0% vs. 30.2%, respectively, *p* = 0.26). Hyperleukocytosis (hazard ratio [HR]: 2.04), extreme hyperleukocytosis (HR: 2.71), age less than 1 year (HR: 3.05), age greater than 10 years (HR: 1.64), and male sex (HR: 1.37) were independently associated with poorer overall survival in childhood ALL patients. Extreme hyperleukocytosis (HR: 2.63) and age less than 1 year (HR: 1.82) were independently associated with poorer overall survival in AML patients. Hyperleukocytosis was associated with adverse survival outcomes in childhood leukemia.

## 1. Introduction

Acute leukemia is the most common malignancy in childhood, accounting for one-third of all malignancies [1,2,3]. Hyperleukocytosis, defined as an initial white blood cell (WBC) count greater than 100 × 10^9^/L, is a serious presenting feature, with reported incidences ranging from 10.2 to 19.2% in childhood acute lymphoblastic leukemia (ALL) patients [4,5,6,7,8,9,10,11,12], and 12.6 to 21.7% in childhood acute myeloid leukemia (AML) patients [4,6,13,14,15,16]. Childhood ALL with hyperleukocytosis has been associated with a number of characteristics that are known to increase the likelihood of adverse outcomes, including age at diagnosis, male gender, T-cell immunophenotype, massive hepatosplenomegaly, and high blood lactate dehydrogenase (LDH) levels [5,7,8,17]. Patients presenting with hyperleukocytosis are at risk for developing early treatment-related complications secondary to leukostasis, including seizures, intracranial bleeding, respiratory problems, coagulopathy, renal failure, and metabolic abnormalities related to tumor lysis syndrome [5,6,7,8,18]. Previous studies have reported that childhood acute leukemia with hyperleukocytosis have poor survival outcomes [5,7,8,15]. However, most of these studies were conducted in developed countries; few have been conducted in developing countries, where treatment outcomes remain worse than in developed countries [19]. This study aimed to determine the clinical course and survival outcomes of childhood acute leukemia with hyperleukocytosis in Thailand, a developing country in South-East Asia.

## 2. Materials and Methods

We retrospectively reviewed the medical records of children aged less than 15 years who had been diagnosed with acute leukemia and had received chemotherapy between January 1998 and December 2017 at the Oncology Clinic, Department of Pediatrics, Faculty of Medicine, Prince of Songkla University, the principal tertiary referral center in southern Thailand. The demographic characteristics, clinical symptoms at diagnosis, initial laboratory investigations, subtype of acute leukemia (ALL or AML), and treatment-related complications were recorded. Diagnosis of subtype was made according to the French–American–British (FAB) classification by morphological examination of bone marrow staining. Three cytochemical stains were used in the diagnostic process: periodic acid-Schiff, peroxidase, and α-naphthyl acetate esterase. After 2000, immunophenotyping of cell surface markers was additionally used to differentiate subtypes (i.e., AML from ALL, and T-cell from B-cell ALL). Prior to 2006, our chemotherapy treatment protocols were based on the Children’s Cancer Study Group’s modified ALL treatment protocols [20,21,22] and the AML treatment protocols of the Berlin–Frankfurt–Munster group [23,24]. In 2006, nationwide standardized protocols for treating childhood leukemia were developed by the Thai Pediatric Oncology Group [21,25,26,27]. Our treatment protocols were based on the national protocols according to the subtype of leukemia and risk stratification of the patient at presentation. In 2014, an updated version of the Thai national protocols according to the subtypes of leukemia and risk stratification of patients at presentation was implemented [28,29]. 

Hyperleukocytosis was defined as a WBC count at initial presentation greater than 100 × 10^9^/L, and extreme hyperleukocytosis was defined as a WBC count at initial presentation greater than 200 × 10^9^/L. Treatment-related complications which occurred during the induction phase of chemotherapy, namely tumor lysis syndrome, seizure, intracranial hemorrhage, acute kidney injury (AKI), septic shock, disseminated intravascular coagulation (DIC), endotracheal tube (ETT) intubation, and intensive care unit (ICU) admission, were recorded. Tumor lysis syndrome was diagnosed according to the Cairo and Bishop criteria, namely, the presence of two or more of the following abnormal conditions: hyperkalemia (serum potassium level ≥ 6 mmol/L), hyperuricemia (serum uric acid level ≥ 8 mmol/L), hyperphosphatemia (serum phosphate level ≥ 6.5 mmol/L), and/or hypocalcemia (serum calcium level < 7 mmol/L) within 3 days before or 7 days after the start of chemotherapy [30]. Acute kidney injury was defined according to the Kidney Disease: Improving Global Outcomes (KDIGO) criteria [31]. Septic shock was defined according to international pediatric sepsis consensus guidelines [32]. Disseminated intravascular coagulation was defined according to the International Society on Thrombosis and Hemostasis (ISTH) scoring system [33].

### Statistical Analysis

Descriptive statistics are presented using mean and standard deviation (SD) or median and interquartile range (IQR) for continuous variables as appropriate. Categorical variables are presented using frequency and percentage. We compared continuous variables between children with and without hyperleukocytosis using Student’s *t*-test for normally distributed variables and the rank-sum test otherwise. Categorical variables were compared using the chi-square or Fisher’s exact tests as appropriate. Overall and event-free survival were compared using log-rank tests and survival distributions were depicted visually with Kaplan–Meier curves. Risk factors for overall survival were analyzed using Cox regression analysis. Variables having a *p* value less than 0.2 from the univariate analysis were included in the initial multivariate Cox regression model for assessment of independent risk factors. The risk factors for overall survival are presented as adjusted hazard ratios (HR) with 95% confidence intervals (CI). R version 4.2.1 was used for all analyses. A *p*-value less than 0.05 was considered significant.

## 3. Results

### 3.1. Study Population

During the 20-year study period, a total of 779 children were diagnosed with acute leukemia in our center. We excluded 39 who had incomplete data and 50 whose parents refused to allow their child to receive standard chemotherapy, leaving a total of 690 children for analysis. Of these, 483 were diagnosed with ALL and 207 with AML. The median (IQR) age at diagnosis of all patients was 55.5 (34.0–105.0) months and the majority (57.5%) were male. The median (IQR) survival time was 4.36 (1.11–12.90) years. The 5-year and 10-year overall survival rates were 52.3% and 47.7%, respectively. Most (77.9%) ALL patients had B-cell subtype. Approximately 60% of ALL patients had cytogenetic studies, and one-fourth of these displayed aberrant cytogenetic findings such as hyperdiploidy, hypodiploidy, and complex karyotype. Cytogenetic studies were performed on almost 70% of AML patients, and around 50% of these showed abnormal cytogenetic findings such as trisomy 21, trisomy 8, monosomy 7, t(8; 21), and complex karyotype.

### 3.2. Hyperleukocytosis in Acute Lymphoblastic Leukemia

Among the 483 ALL patients, 80 (16.6%) had hyperleukocytosis at initial presentation and 40 (8.3%) had extreme hyperleukocytosis. Comparisons of demographic characteristics and outcomes between children with and without hyperleukocytosis are presented in Table 1. Those with hyperleukocytosis were significantly older than those without hyperleukocytosis (91 months vs. 52 months, *p* = 0.013). A higher proportion with hyperleukocytosis had T-cell subtype compared to those without hyperleukocytosis (31.2% vs. 9.4%, *p* = 0.04). Mediastinal mass, hepatomegaly and splenomegaly were significantly more common in the hyperleukocytosis group (*p* < 0.05). Those with hyperleukocytosis had a significantly lower average platelet count and a higher percentage of peripheral blast cells (*p* < 0.05). Average serum calcium, phosphorus and LDH levels were significantly lower, whereas the average uric acid level was significantly higher, in the hyperleukocytosis group (*p* < 0.05). Regarding treatment-related complications during the induction phase of chemotherapy, a significantly higher proportion of those with hyperleukocytosis developed tumor lysis syndrome, seizure, acute kidney injury, septic shock, and disseminated intravascular coagulation required endotracheal tube intubation and intensive care unit admission. Those with hyperleukocytosis also had significantly lower remission rates of induction and higher mortality rates compared with those without hyperleukocytosis (*p* = 0.011 and *p* < 0.001, respectively).

### 3.3. Hyperleukocytosis in Acute Myeloid Leukemia

Among the 207 AML patients, 42 (20.3%) had hyperleukocytosis and 22 (10.6%) had extreme hyperleukocytosis at initial presentation. Comparisons of the demographic characteristics and outcomes between those with and without hyperleukocytosis are presented in Table 2. Those with hyperleukocytosis were significantly older than those without hyperleukocytosis (137 months vs. 58 months, *p* = 0.021). Those with hyperleukocytosis had significantly more fever, splenomegaly and lymphadenopathy than those without hyperleukocytosis (*p* < 0.05) and a significantly higher percentage of peripheral blast cells (*p* < 0.05). The average serum calcium and phosphorus levels were significantly lower, whereas the average uric acid level was significantly higher, in the hyperleukocytosis group (*p* < 0.05). The rates of treatment-related complications during the induction phase of chemotherapy were not significantly different between those with and without hyperleukocytosis, except for intracranial hemorrhage, which was only observed in a small number of patients with hyperleukocytosis. Treatment outcomes (remission, relapse and mortality) were not significantly different between the two groups.

### 3.4. Survival Outcomes in Acute Lymphoblastic Leukemia

The median survival time for the 483 ALL patients was 4.32 (3.31–9.27) years, and the 5-year and 10-year overall survival rates were 48.9% and 44.9%, respectively. The median survival times for the patients with hyperleukocytosis and those without hyperleukocytosis were 1.84 (1.51–4.28) years and 23.2 (22.8–∞) years, respectively. The Kaplan–Meier curves comparing overall survival (OS) between ALL patients with and without hyperleukocytosis are presented in Figure 1. The 5-year OS and EFS rates for those with hyperleukocytosis were both significantly lower than for those without (OS, 37.2% vs. 67.8%, respectively, *p* < 0.0001, and EFS, 33.7% vs. 59.1%, respectively, *p* < 0.0001).

In the multivariate Cox regression analysis, risk factors for overall survival among the ALL patients are shown in Table 3. The independent risk factors for poorer overall survival were age, sex and WBC counts at initial presentation. Age less than 1 year and age greater than 10 years had hazard ratios of 3.05 (95% CI: 1.57–5.96) and 1.64 (95% CI: 1.51–2.32), respectively, when compared to those aged between 1 and 9 years (*p* = 0.001 and *p* = 0.006, respectively). Males had a hazard ratio of 1.37 (95% CI: 1.05–1.79) when compared to females (*p* = 0.021). Hyperleukocytosis and extreme hyperleukocytosis had hazard ratios of 2.04 (95% CI: 1.33–3.14) and 2.71 (95% CI: 1.74–4.21) compared with those who had an initial WBC count less than 50 × 10^9^/L (*p* = 0.001 and *p* <0.001, respectively). Patients who had an initial WBC count of 50 to 100 × 10^9^/L had a hazard ratio of 1.59 (95% CI: 1.07–2.36) compared with those who had an initial WBC count less than 50 × 10^9^/L (*p* = 0.022). 

### 3.5. Survival Outcomes in Acute Myeloid Leukemia

The median survival times for the 42 AML patients with hyperleukocytosis and 165 AML patients without hyperleukocytosis were 1.11 (0.87–1.76) years and 0.83 (0.33–2.03) years, respectively. The Kaplan–Meier curves showing overall survival between children with and without hyperleukocytosis are presented in Figure 2. The OS and EFS rates were lower for those with hyperleukocytosis compared with those without hyperleukocytosis, but the differences were not statistically significant (*p* = 0.26 and 0.29, respectively). The 5-year OS rate for those with hyperleukocytosis was 19.0% compared with 30.2% for those without. The 5-year EFS rate for those with hyperleukocytosis was 19.0% compared with 27.7% for those without.

In the multivariate Cox regression analysis, risk factors for overall survival in childhood AML patients are shown in Table 4. The independent risk factors for poorer overall survival were extreme hyperleukocytosis at initial presentation and age. Extreme hyperleukocytosis had a hazard ratio of 2.63 (95% CI: 1.56–4.43) compared with those who had initial WBC count less than 50 × 10^9^/L (*p* < 0.001), while those aged less than 1 year had a hazard ratio of 1.82 (95% CI: 1.00–3.32) when compared to those aged between 1 and 9 years (*p* = 0.049).

## 4. Discussion

Our long-term study covered a 20-year period and included a total of 690 childhood acute leukemia patients, consisting of 483 ALL and 207 AML patients. We found that 16.6% of ALL patients and 20.3% of AML patients had hyperleukocytosis at initial presentation. Previous studies have reported incidences of hyperleukocytosis at initial presentation of childhood ALL patients ranging from 10.2 to 19.2% [4,5,6,7,8,9,10] and in childhood AML patients ranging from 12.6 to 21.7% [4,6,13,14,15,16]. However, the majority of these studies were conducted in developed countries and there are limited data from developing countries. Two studies from India reported that 14.6% and 16.7% of childhood ALL patients had hyperleukocytosis at presentation [11,12]. However, these studies had limited data on the clinical characteristics and outcomes of their hyperleukocytosis patients. Our study found that the incidence of hyperleukocytosis in childhood leukemia patients was comparable to that of previous studies.

Hyperleukocytosis in childhood ALL has been associated with a variety of clinical characteristics [5,7,8]. Eguiguren et al. reported that hyperleukocytosis was significantly associated with age less than 1 year, T-cell immunophenotype, mediastinal mass, massive hepatosplenomegaly, and elevated LDH [5]. Kong et al. retrospectively reviewed 104 childhood ALL patients and found that age 1–10 years, male gender, T-cell immunophenotype, and massive splenomegaly were associated with extreme hyperleukocytosis (WBC > 200 × 10^9^/L) [7]. A later multicenter study by Park et al. reported that children aged ≥ 10 years were more likely to develop extreme hyperleukocytosis [8]. Nguyen et al. reported that childhood ALL with extreme hyperleukocytosis was associated with age younger than 1 or older than 9 years of age, male, T-cell immunophenotype, central nervous system involvement, elevated LDH, and elevated uric acid [18]. Similarly, our study found that hyperleukocytosis in childhood ALL was associated with older age, T-cell immunophenotype, mediastinal mass, hepatomegaly, and splenomegaly. Regarding childhood AML patients, data on the associated factors that may enhance the risk of hyperleukocytosis are limited. Sung et al. reported that hyperleukocytosis in childhood AML patients was associated with an age less than 1 year, FAB M1, M4, and M5, and certain chromosomal abnormalities [14]. In contrast, our study found that older age, fever, lymphadenopathy, and splenomegaly were clinical features significantly associated with hyperleukocytosis in childhood AML. However, factors that increase the risk of hyperleukocytosis in childhood ALL and AML remain inconclusive. Further studies, including cytogenetic ones, with larger sample sizes, multicenter designs, and diverse populations are needed.

Earlier studies found that early treatment-related complications were more common in childhood ALL patients who had hyperleukocytosis at initial presentation. Bunin et al. reported that metabolic complications occurred in 13.7% of 161 childhood ALL patients with hyperleukocytosis [4]. Alba et al. found that among 84 childhood ALL cases with hyperleukocytosis, 19.0% had respiratory complications followed by neurological complications and renal dysfunction in 7.1% and 6.0% of their cases, respectively [6]. Kong et al. reported that metabolic complications occurred in 40% of 20 childhood ALL patients with hyperleukocytosis, followed by neurological and respiratory complications in 20% and 10%, respectively [7]. Park et al. found that among 72 childhood ALL cases with hyperleukocytosis, metabolic complications developed in 59.7%, followed by respiratory and neurologic complications in 11.1% and 9.7%, respectively [8]. Similarly, our study found that the rates of early treatment-related complications among childhood ALL patients with hyperleukocytosis were comparable to previous studies, namely septic shock (32.5%), metabolic complications related to tumor lysis syndrome (30.0%), disseminated intravascular coagulation (26.3%), required endotracheal tube intubation (17.5%), acute kidney injury (11.3%), seizure (11.3%), and intracranial hemorrhage (3.8%). The varying incidences of treatment-related complications reported in the literature might be due, at least partly, to differences in the definitions among the studies. Nevertheless, our study also demonstrated that early treatment-related complications were significantly higher in childhood ALL patients with hyperleukocytosis compared with those without.

In a comparable pattern to the childhood ALL population, hyperleukocytosis in children with AML has been associated with early complications. Bunin et al. reported that metabolic complications, acute renal failure, and intracerebral hemorrhage occurred in 5.5%, 5.5%, and 11% of 73 childhood AML patients with hyperleukocytosis, respectively. In another study serious hemorrhagic complications were reported in significantly more childhood AML patients with hyperleukocytosis than in childhood ALL patients with hyperleukocytosis (19% vs. 2.5%) [4]. Inaba et al. found that among 106 childhood AML patients with hyperleukocytosis, early complications developed in 42.5%, including respiratory, neurological and renal complications at rates of 18.9%, 16%, and 15%, respectively [15]. Sung et al. reported that a high WBC count in childhood AML cases was significantly associated with cerebral nervous system ischemia or hemorrhage, hypoxia, and pulmonary hemorrhage [14]. Alba et al. found that among 18 childhood ALL cases with hyperleukocytosis, respiratory, neurological, and renal complications developed in 22.2%, 22.2%, and 11.1% of their cases, respectively [6]. Our study observed that the rates of early treatment-related complications among childhood AML patients with hyperleukocytosis were less than in previous studies. However, the incidence of early complications may vary depending on the definitions applied and the limited sample sizes of childhood AML cases with hyperleukocytosis in the literature. Further multicenter prospective studies are required.

Our study found that the survival outcomes for childhood leukemia patients in the entire cohort were comparable to previous studies from developing countries but were lower than studies conducted in developed countries [34,35]. The subgroup of childhood ALL cases with hyperleukocytosis has been associated with poor survival outcomes. Eguiguren et al. reported that the 4-year EFS rate of 64 childhood ALL patients with hyperleukocytosis was significantly lower than in those without (52% vs. 75%, respectively) [5]. Kong et al. reported that the 3-year OS and EFS rates of 104 childhood ALL cases with hyperleukocytosis were 81.2% and 75%, respectively [7]. Park et al. found that the 10-year OS and EFS rates among 72 childhood ALL cases with hyperleukocytosis were 82.6% and 78.3%, respectively [8]. Maurer et al. reported that the EFS for 106 childhood ALL patients with an initial WBC count > 200 × 10^9^/L was 55% at 3 years [36]. However, most of these studies were conducted in developed countries, and there have been few studies from developing countries. Our study found lower survival rates for childhood ALL with hyperleukocytosis (5-year EFS rate of 33.7%) than all studies from developed countries, and that children with hyperleukocytosis had poorer survival rates than children without hyperleukocytosis. When focusing on the subgroup of childhood AML cases with hyperleukocytosis, a study by Inaba et al. reported that those with hyperleukocytosis had a lower 10-year EFS rate compared with those without, but the difference in the 10-year OS rates was not significant. Similarly, our study found that the OS and EFS rates were lower for the AML patients with hyperleukocytosis compared with those without, however the difference was not statistically significant.

Several factors, including age, male sex, WBC count at diagnosis, central nervous system involvement, T-cell immunophenotype, race, and cytogenetics have been considered as adverse prognostic factors in childhood ALL [37,38,39,40]. Age and WBC count at initial diagnosis were factors used by The National Cancer Institute (NCI) risk classification to separate high and standard risk groups [41]. Eguiguren et al. reported that only leukocyte count was identified as a significant prognostic factor for EFS [5]. A study by Kong et al. found associations between EFS and WBC count, age, splenomegaly, mediastinal mass, immunophenotype, and chromosomal abnormalities. However, multivariate analysis was not performed [7]. Maurer et al. reported that a WBC count at diagnosis greater than 600 × 10^9^/L and massive splenomegaly were the only two significant adverse prognostic factors for survival on multivariate analysis with hazard ratios of 2.60 and 2.42, respectively [36]. Our study found that on multivariate analysis, age, male sex, and WBC count at initial presentation were independently associated with overall survival in childhood ALL patients, consistent with results from other studies.

Several factors associated with early mortality in the AML population have been reported in the literature. Bunin et al. found that early mortality in childhood AML cases with hyperleukocytosis (23.3%) was significantly higher than in childhood ALL cases with hyperleukocytosis (5%) [4]. Abla et al. reported the rate of early death was 16.7% in 18 childhood AML patients with hyperleukocytosis compared with 1.2% in 634 childhood ALL patients. They also found that early death due to hyperleukocytosis in childhood leukemia was associated with neurologic complications, AML diagnosis, and initial coagulopathy [6]. Creutzig et al. found that early mortality from hemorrhage or leukostasis occurred in 30.8% of 65 childhood AML patients with hyperleukocytosis and the risk factors for early death were FAB M5, hyperleukocytosis and the presence of extramedullary organ involvement [42]. Nevertheless, the literature on the factors that affect overall survival in the population of children with AML is still sparse. Our study found that extreme hyperleukocytosis and age less than 1 year were independent associated factors for overall survival in childhood AML patients. Our opinion is that treatment recommendations for hyperleukocytosis in both AML and ALL patients should be modified to be more intensive in those who have identified risk factors to reduce complications secondary to leukostasis while being aware of the risk of developing consequences resulting from treatment by close monitoring and providing intensive care.

There is no standard treatment guideline for hyperleukocytosis in Thailand. The current practice focuses on preventing and reducing complications secondary to leukostasis by providing intensive supportive care and prompt cytoreduction by means of hydration, urine alkalinization, allopurinol administration, early chemotherapy induction, and occasionally leukapheresis. In our center, patients with hyperleukocytosis were managed by hydration, urine alkalinization, allopurinol administration, and early chemotherapy induction. Less than 10% of the ALL patients with hyperleukocytosis received pretreatment with corticosteroids. None of the AML patients with hyperleukocytosis received pretreatment with cytarabine. Leukapheresis was only performed on a small number of patients (3.5%) since the efficacy is still controversial and the procedure is constrained by age, availability, the requirement for central venous access, and the potential for adverse effects. Due to the small proportion of patients who underwent leukapheresis, our study was restricted to evaluating the efficacy of leukapheresis in patients with hyperleukocytosis. Further multicenter studies involving a larger number of patients are needed.

The notable strength of our study was that it included a large number of patients with a long follow-up period. However, the study also had some limitations. First, this study covered a 20-year period during which there were significant advancements in both treatment and supportive care for cancer patients, and these advances would undoubtedly have had some impact on the outcomes which we were unable to assess in this study. Second, the statistical analysis was also impacted by factors such as the multiple periods of diagnosis where there may have been differences in classification of ALL (i.e., from FAB to T-cell or B-cell ALL) and the method of diagnosis, which in later years additionally used immunophenotyping of cell markers. Third, our study had a limited number of cytogenetic studies and lack of molecular and minimal residual disease (MRD) studies.

## 5. Conclusions

Hyperleukocytosis in childhood leukemia was associated with higher rates of morbidity and worse survival outcomes. Hyperleukocytosis, extreme hyperleukocytosis, age less than 1 year, age greater than 10 years, and male sex were independently associated with a poorer overall survival in childhood ALL patients. Extreme hyperleukocytosis and age less than 1 year were independently associated with poorer overall survival in childhood AML patients.

## Figures and Tables

**Figure 1 cancers-15-03072-f001:**
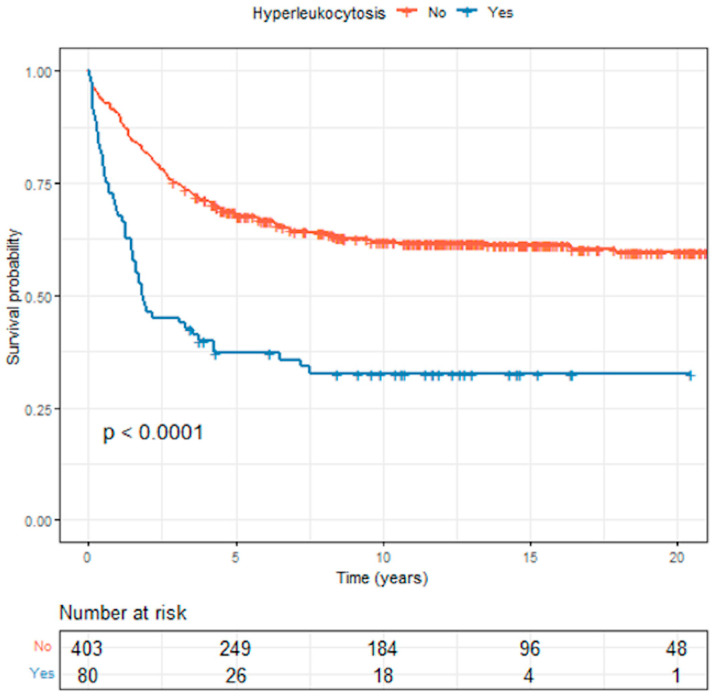
Comparison of overall survival between childhood acute lymphoblastic leukemia patients with and without hyperleukocytosis.

**Figure 2 cancers-15-03072-f002:**
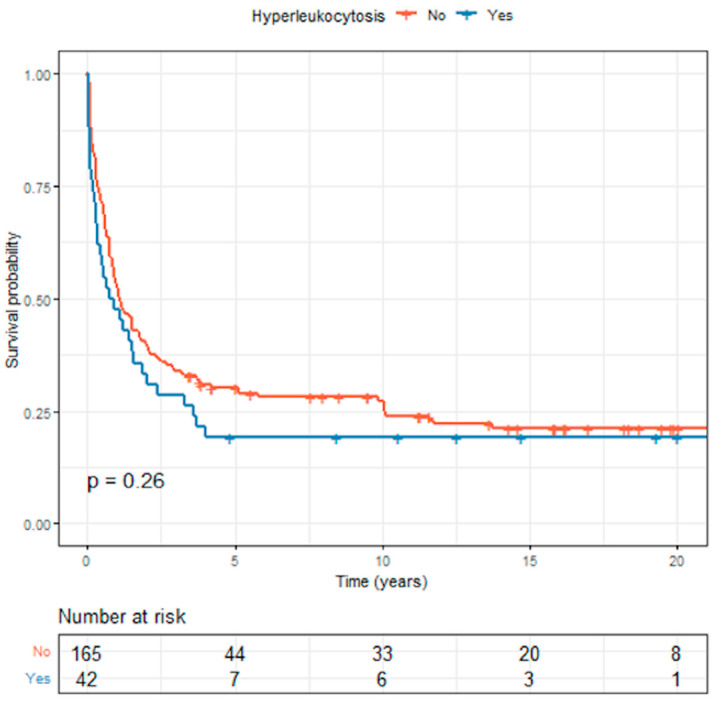
Comparison of overall survival between childhood acute myeloid leukemia patients with and without hyperleukocytosis.

**Table 1 cancers-15-03072-t001:** Comparison of demographic characteristics and outcomes between childhood acute lymphoblastic leukemia patients with and without hyperleukocytosis.

Variables	Total (N = 483)	Hyperleukocytosis (N = 80)	No Hyperleukocytosis (N = 403)	*p* Value
Demographic and clinical characteristics, n (%)				
Age (months), median (IQR)	54.0 (36.0–97.0)	91.0 (34.0–137.0)	52.0 (36.0–86.0)	0.013
Sex				0.171
Male	272 (56.3)	39 (48.8)	233 (57.8)	
Female	211 (43.7)	41 (51.2)	170 (42.2)	
Immunophenotype				<0.001
T-cell	63 (13.0)	25 (31.2)	38 (9.4)	
B-cell	376 (77.9)	51 (63.8)	325 (80.7)	
FAB classification	44 (9.1)	4 (5.0)	40 (9.9)	
Fever	360 (74.5)	62 (77.5)	298 (73.9)	0.599
Mediastinal mass	23 (4.8)	8 (10.0)	15 (3.7)	0.037
Hepatomegaly	448 (92.8)	80 (100.0)	368 (91.3)	0.012
Splenomegaly	332 (68.7)	72 (90.0)	260 (64.5)	<0.001
Laboratory parameters, mean (SD)				
Hemoglobin (g/dL)	7.5 (3.5)	7.0 (2.7)	7.6 (3.7)	0.223
Platelet count (×10^9^/L)	75.5 (92.3)	38.0 (29.9)	82.9 (98.5)	<0.001
Blast cells (%)	50.2 (34.8)	90.4 (11.5)	42.2 (32.2)	<0.001
Calcium (mmol/L)	9.6 (1.1)	9.3 (0.9)	9.6 (1.1)	0.018
Phosphorus (mmol/L)	4.8 (1.3)	4.1 (1.5)	5 (1.3)	<0.001
Uric acid (mmol/L)	6.1 (3.5)	7.5 (4.1)	5.9 (3.3)	<0.001
Lactate dehydrogenase (U/L)	2776.3 (5339.4)	4971.5 (7105.8)	2350.3 (4820.7)	<0.001
Treatment-related complications, n (%)				
Tumor lysis syndrome	67 (13.9)	24 (30.0)	43 (10.7)	<0.001
Seizure	21 (4.3)	9 (11.3)	12 (3.0)	0.003
Intracranial hemorrhage	6 (1.2)	3 (3.8)	3 (0.7)	0.056
Acute kidney injury	27 (5.6)	9 (11.3)	18 (4.5)	0.025
Septic shock	106 (21.9)	26 (32.5)	80 (19.9)	0.009
DIC	65 (13.5)	21 (26.3)	44 (10.9)	<0.001
ETT intubation	38 (7.9)	14 (17.5)	24 (6.0)	<0.001
ICU admission	50 (10.4)	22 (27.5)	28 (6.9)	<0.001
Treatment outcomes, n (%)				
Induction of remission	428 (96.8)	63 (91.3)	365 (97.9)	0.011
Relapse	162 (33.5)	31 (38.8)	131 (32.5)	0.342
Mortality	207 (42.9)	53 (66.3)	154 (38.2)	<0.001

Values are expressed as n (%), mean (SD), or median (IQR). DIC, disseminated intravascular coagulopathy; ETT, endotracheal tube; FAB, French–American–British classification system; ICU, intensive care unit.

**Table 2 cancers-15-03072-t002:** Comparison of demographic characteristics and outcomes between childhood acute myeloid leukemia patients with and without hyperleukocytosis.

Variables	Total (N = 207)	Hyperleukocytosis (N = 42)	No Hyperleukocytosis (N = 165)	*p* Value
Demographic and clinical characteristics, n (%)				
Age (months), median (IQR)	65.0 (25.5–133.5)	137.0 (34.0–150.5)	58.0 (25.0–116.0)	0.021
Sex				0.268
Male	125 (60.4)	29 (69.0)	96 (58.2)	
Female	82 (39.6)	13 (31.0)	69 (41.8)	
Fever	157 (75.8)	39 (92.9)	118 (71.5)	0.007
Hepatomegaly	165 (79.7)	37 (88.1)	128 (77.6)	0.194
Splenomegaly	119 (57.5)	31 (73.8)	88 (53.3)	0.026
Lymphadenopathy	142 (68.6)	35 (83.3)	107 (64.8)	0.034
Laboratory parameters, mean (SD)				
Hemoglobin (g/dL)	7.3 (2.2)	7.1 (1.6)	7.3 (2.3)	0.585
Platelet count (×10^9^/L)	64.1 (93.1)	50.0 (35.3)	67.7 (102.5)	0.272
Blast cells (%)	43.4 (34.7)	80.5 (24.7)	33.9 (30.3)	<0.001
Calcium (mmol/L)	9.3 (0.7)	9.0 (0.9)	9.3 (0.7)	0.021
Phosphorus (mmol/L)	4.6 (1.2)	4.1 (1.4)	4.8 (1.0)	<0.001
Uric acid (mmol/L)	5.0 (2.1)	6.0 (2.6)	4.8 (1.9)	<0.001
Lactate dehydrogenase (U/L)	2035.6 (2210.0)	2619.7 (2425.5)	1888.6 (2135.3)	0.058
Treatment-related complications, n (%)				
Tumor lysis syndrome	11 (5.3)	4 (9.5)	7 (4.2)	0.276
Seizure	8 (3.9)	1 (2.4)	7 (4.2)	1
Intracranial hemorrhage	2 (1.0)	2 (4.8)	0 (0)	0.037
Acute kidney injury	14 (6.8)	2 (4.8)	12 (7.3)	1
Septic shock	59 (28.5)	11 (26.2)	48 (29.1)	1
DIC	49 (23.7)	13 (31.0)	36 (21.8)	0.219
ETT intubation	41 (19.8)	8 (19.0)	33 (20.0)	1
ICU admission	46 (22.2)	11 (26.2)	35 (21.2)	0.504
Treatment outcomes, n (%)				
Induction of remission	109 (67.3)	19 (59.3)	90 (69.2)	0.289
Relapse	63 (30.4)	12 (28.6)	51 (30.9)	0.915
Mortality	160 (77.3)	34 (81.0)	126 (76.4)	0.669

Values are expressed as n (%), mean (SD), or median (IQR). DIC, disseminated intravascular coagulopathy; ETT, endotracheal tube; ICU, intensive care unit.

**Table 3 cancers-15-03072-t003:** Multivariate analysis results showing independent risk factors for overall survival in childhood acute lymphoblastic leukemia patients.

Risk Factor	Crude HR (95% CI)	Adjusted HR (95% CI)	*p* Value
Age (years)			
1–9	Reference	Reference	
<1	3.51 (1.79–6.89)	3.05 (1.57–5.96)	0.001
≥10	1.80 (1.29–2.52)	1.64 (1.15–2.32)	0.006
Male sex	1.11 (0.84–1.47)	1.37 (1.05–1.79)	0.021
Immunophenotype			
T-cell	Reference	Reference	
B-cell	0.79 (0.55–1.15)	1.37 (0.90–2.08)	0.14
FAB classification	0.56 (0.31–1.01)	0.85 (0.46–1.56)	0.6
Initial WBC count (×10^9^/L)			
<50	Reference	Reference	
50–<100	1.47 (0.97–2.23)	1.59 (1.07–2.36)	0.022
≥100–<200	2.45 (1.60–4.61)	2.04 (1.33–3.14)	<0.001
≥200	3.04 (2.00–4.61)	2.71 (1.74–4.21)	<0.001
Early complication	1.16 (0.79–1.71)	0.88 (0.62–1.26)	0.5

HR, hazard ratio; CI, confidence interval; FAB, French–American–British classification system; WBC, white blood cell.

**Table 4 cancers-15-03072-t004:** Multivariate analysis results showing independent risk factors for overall survival in childhood acute myeloid leukemia patients.

Risk Factor	Crude HR (95% CI)	Adjusted HR (95% CI)	*p* Value
Age (years)			
1–9	Reference	Reference	
<1	1.43 (0.80–2.55)	1.82 (1.00–3.32)	0.049
≥10	0.84 (0.59–1.20)	0.70 (0.48–1.02)	0.064
Male sex	1.06 (0.77–1.47)	1.03 (0.74–1.42)	0.9
Initial WBC count (×10^9^/L)			
<50	Reference	Reference	
50–<100	1.13 (0.70–1.82)	1.25 (0.76–2.03)	0.4
≥100–<200	0.83 (0.48–1.43)	0.81 (0.46–1.44)	0.5
≥200	2.16 (1.32–3.53)	2.63 (1.56–4.43)	<0.001
Early complication	0.83 (0.50–1.40)	0.89 (0.53–1.50)	0.7

HR, hazard ratio; CI, confidence interval; WBC, white blood cell.

## Data Availability

The datasets generated and analyzed during the current study are available from the corresponding author on reasonable request.

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
