# Peer review of "Hyperleukocytosis in Childhood Acute Leukemia: Early Complications and Survival Outcomes"

_cancers, 2023, doi:10.3390/cancers15123072_

Round 1
Reviewer 1 Report
1. The authors state that "Those with hyperleukocytosis also had significantly lower remission rates of induction and higher mortality rates compared with those without hyperleukocytosis (p = 141 0.01 and p <0.01, respectively)," but I could not find the results in the table. Please provide early mortality rates for patients with and without hyperleukocytosis.
2. Please indicate whether T cell phenotype was an independent factor for survival in the multivariate analysis.
3. In patients with ALL, was hyperleukocytosis associated with survival in B-cell ALL and in T-cell ALL patients?
4. If available, please provide any cytogenetic data.
5. Please describe how hyperleukocytosis is managed in Thailand. For example, do you use steroids or leukapheresis in ALL patients? Do you use cytarabine or leukapheresis in AML patients?
6. If uniform guidelines for the management of hyperleukocytosis do not exist, please provide recommendations that would be applicable to your country.
Author Response
Reviewer Comments:
Reviewer #1:
Comments to the Author
1. The authors state that "Those with hyperleukocytosis also had significantly lower remission rates of induction and higher mortality rates compared with those without hyperleukocytosis (p = 141 0.01 and p <0.01, respectively)," but I could not find the results in the table. Please provide early mortality rates for patients with and without hyperleukocytosis.
Response: Thank you for your comments. We have revised the manuscript accordingly.
Page 3, line 145: "Those with hyperleukocytosis also had significantly lower remission rates of induction and higher mortality rates compared with those without hyperleukocytosis (p = 0.011 and p <0.001, respectively)."
Variables |
Total (N=483) |
Hyperleukocytosis (N=80) |
No hyperleukocytosis (N=403) |
P value |
Treatment outcomes, n (%) |
||||
Induction of remission |
428 (96.8) |
63 (91.3) |
365 (97.9) |
0.011 |
Relapse |
162 (33.5) |
31 (38.8) |
131 (32.5) |
0.342 |
Mortality |
207 (42.9) |
53 (66.3) |
154 (38.2) |
< 0.001 |
2. Please indicate whether T cell phenotype was an independent factor for survival in the multivariate analysis.
3. In patients with ALL, was hyperleukocytosis associated with survival in B-cell ALL and in T-cell ALL patients?
Response: By adding immunophenotype to a multivariate model, the results changed. Therefore, we have revised the manuscript as follows:
Table 3. Multivariate analysis results showing independent risk factors for overall survival in childhood acute lymphoblastic leukemia patients
Risk factor |
Crude HR |
Adjusted HR |
P value |
Age (years) |
|
|
|
1 – 9 |
Reference |
Reference |
|
< 1 |
3.51 (1.79-6.89) |
3.05 (1.57-5.96) |
0.001 |
≥ 10 |
1.80 (1.29-2.52) |
1.64 (1.15-2.32) |
0.006 |
Male sex |
1.11 (0.84-1.47) |
1.37 (1.05-1.79) |
0.021 |
Immunophenotype |
|
|
|
T-cell |
Reference |
Reference |
|
B-cell |
0.79 (0.55–1.15) |
1.37 (0.90-2.08) |
0.14 |
FAB classification |
0.56 (0.31–1.01) |
0.85 (0.46-1.56) |
0.6 |
Initial WBC count (×109/L) |
|
|
|
< 50 |
Reference |
Reference |
|
50 – < 100 |
1.47 (0.97-2.23) |
1.59 (1.07-2.36) |
0.022 |
≥ 100 – < 200 |
2.45 (1.60-4.61) |
2.04 (1.33-3.14) |
<0.001 |
≥ 200 |
3.04 (2.00-4.61) |
2.71 (1.74-4.21) |
<0.001 |
Early complication |
1.16 (0.79-1.71) |
0.88 (0.62-1.26) |
0.5 |
Page 1, line 15: "Hyperleukocytosis, extreme hyperleukocytosis (WBC >200×109/L), age less than 1 year, age greater than 10 years, and male sex were independently associated with overall survival in the ALL group while extreme hyperleukocytosis and age less than 1 year were independently associated with overall survival in the AML group."
Page 1, line 35: "Hyperleukocytosis (hazard ratio [HR]: 2.04), extreme hyperleukocytosis (HR: 2.71), age less than 1 year (HR: 3.05), age greater than 10 years (HR: 1.64), and male sex (HR: 1.37) were independently associated with poorer overall survival in childhood ALL patients."
Page 6, line 185: "In the multivariate Cox regression analysis, risk factors for overall survival among the ALL patients are shown in Table 3. The independent risk factors for poorer overall survival were age, sex and WBC counts at initial presentation. Age less than 1 year and age greater than 10 years had hazard ratios of 3.05 (95% CI: 1.57 - 5.96) and 1.64 (95% CI: 1.51 - 2.32), respectively, when compared to those aged between 1 and 9 years (p = 0.001 and p = 0.006, respectively). Males had a hazard ratio of 1.37 (95% CI: 1.05 - 1.79) when compared to females (p = 0.021). Hyperleukocytosis and extreme hyperleukocytosis had hazard ratios of 2.04 (95% CI: 1.33 - 3.14) and 2.71 (95% CI: 1.74 - 4.21) compared with those who had an initial WBC count less than 50×109/L (p = 0.001 and p <0.001, respectively). Patients who had an initial WBC count of 50 to 100×109/L had a hazard ratio of 1.59 (95% CI: 1.07 - 2.36) compared with those who had an initial WBC count less than 50×109/L (p = 0.022)."
Page 10, line 328: "Our study found that on multivariate analysis, age, male sex, and WBC count at initial presentation were independently associated with overall survival in childhood ALL patients, consistent with results from other studies."
Page 11, line 375: "Hyperleukocytosis, extreme hyperleukocytosis, age less than 1 year, age greater than 10 years, and male sex were independently associated with a poorer overall survival in childhood ALL patients."
4. If available, please provide any cytogenetic data.
Response: Thank you for your comments. We have revised the manuscript accordingly.
Page 3, line 124: "Approximately 60% of ALL patients had cytogenetic studies, and one-fourth of these displayed aberrant cytogenetic findings such as hyperdiploidy, hypodiploidy, and complex karyotype. Cytogenetic studies was performed on almost 70% of AML patients, and around 50% of these showed abnormal cytogenetic findings such as trisomy 21, trisomy 8, monosomy 7, t(8;21), and complex karyotype."
5. Please describe how hyperleukocytosis is managed in Thailand. For example, do you use steroids or leukapheresis in ALL patients? Do you use cytarabine or leukapheresis in AML patients?
6. If uniform guidelines for the management of hyperleukocytosis do not exist, please provide recommendations that would be applicable to your country.
Response: We have added more details and revised the manuscript.
Page 11, line 349: "There is no standard treatment guideline for hyperleukocytosis in Thailand. The current practice focuses on preventing and reducing complications secondary to leukostasis by providing intensive supportive care and prompt cytoreduction by means of hydration, urine alkalinization, allopurinol administration, early chemotherapy induction, and occasionally leukapheresis. In our center, patients with hyperleukocytosis were managed by hydration, urine alkalinization, allopurinol administration, and early chemotherapy induction. Less than 10% of the ALL patients with hyperleukocytosis received pretreatment with corticosteroids. None of the AML patients with hyperleukocytosis received pretreatment with cytarabine. Leukapheresis was only performed on a small number of patients (3.5%) since the efficacy is still controversial and the procedure is constrained by age, availability, the requirement for central venous access, and the potential for adverse effects. Due to the small proportion of patients who underwent leukapheresis, our study was restricted to evaluating the efficacy of leukapheresis in patients with hyperleukocytosis. Further multicenter studies involving a larger number of patients are needed."
Page 11, line 344: "Our opinion is that treatment recommendations for hyperleukocytosis in both AML and ALL patients should be modified to be more intensive in those who have identified risk factors to reduce complications secondary to leukostasis while being aware of the risk of developing consequences resulting from treatment by close monitoring and providing intensive care."

Reviewer 2 Report
The study estimates the incidence of hyperleukocytosis among childhood acute lymphoblastic leukemia and acute myeloid leukemia, identifies associated factors, and impact on survival, using a relatively large cohort. The statistical analysis has rigor.
1. One of the strengths of the study is said to be the long follow up period. It would be useful to indicate the median follow up period for this cohort.
2. Is there any information on how hyperleukocytosis was managed in these patients other than standard chemotherapy? Any specific cytoreduction, apheresis? While it might not be with the scope of the study, if this information is present, comparing their efficacies would be great. If it isn't readily available, the study is still relevant as is.
3. The lack of molecular and cytogenetic data as a limitation is mentioned, but is expected in lesser resourced areas. Was any form of MRD studies however performed (by flow cytometry for instance)?
4. In the simple summary, there is the suggestion that current treatment protocols have to be modified. A discussion on suggested modifications, and also the specific approaches to managing hyperleukocytosis in this setting, would be necessary.
5. In the association of hyperleukocytosis with mediastinal mass, do you think it is through the association with T-cell immunophenotype or you see same association in B-ALL.
5. The study in some ways reinforce the idea behind the NCI risk criteria. While your cohort though sizable is still small, what do you think is the reason why the 50k-100k WBC group in ALL did not sure worse OS compared to reference. The threshold for the NCI is 50k.
6. In the table 2, I think what is meant is induction of remission, and not remission of induction.
Author Response
Reviewer #2:
Comments to the Author
The study estimates the incidence of hyperleukocytosis among childhood acute lymphoblastic leukemia and acute myeloid leukemia, identifies associated factors, and impact on survival, using a relatively large cohort. The statistical analysis has rigor.
Response: Thank you for your comments.
1. One of the strengths of the study is said to be the long follow up period. It would be useful to indicate the median follow up period for this cohort.
Response: We have revised the manuscript accordingly.
Page 3, line 122: "The median (IQR) survival time was 4.36 (1.11-12.9) years."
2. Is there any information on how hyperleukocytosis was managed in these patients other than standard chemotherapy? Any specific cytoreduction, apheresis? While it might not be with the scope of the study, if this information is present, comparing their efficacies would be great. If it isn't readily available, the study is still relevant as is.
Response: We have added more details and revised the manuscript.
Page 11, line 349: "There is no standard treatment guideline for hyperleukocytosis in Thailand. The current practice focuses on preventing and reducing complications secondary to leukostasis by providing intensive supportive care and prompt cytoreduction by means of hydration, urine alkalinization, allopurinol administration, early chemotherapy induction, and occasionally leukapheresis. In our center, patients with hyperleukocytosis were managed by hydration, urine alkalinization, allopurinol administration, and early chemotherapy induction. Less than 10% of the ALL patients with hyperleukocytosis received pretreatment with corticosteroids. None of the AML patients with hyperleukocytosis received pretreatment with cytarabine. Leukapheresis was only performed on a small number of patients (3.5%) since the efficacy is still controversial and the procedure is constrained by age, availability, the requirement for central venous access, and the potential for adverse effects. Due to the small proportion of patients who underwent leukapheresis, our study was restricted to evaluating the efficacy of leukapheresis in patients with hyperleukocytosis. Further multicenter studies involving a larger number of patients are needed."
4. The lack of molecular and cytogenetic data as a limitation is mentioned, but is expected in lesser resourced areas. Was any form of MRD studies however performed (by flow cytometry for instance)?
Response: We have added more details about cytogenetic data. We agree that minimal residual disease (MRD) studies may provide more convincing results. Unfortunately, there was a lack of data concerning this point in most of the patients in our study. We have therefore added this limitation to the manuscript.
Page 3, line 124: "Approximately 60% of ALL patients had cytogenetic studies, and one-fourth of these displayed aberrant cytogenetic findings such as hyperdiploidy, hypodiploidy, and complex karyotype. Cytogenetic studies was performed on almost 70% of AML patients, and around 50% of these showed abnormal cytogenetic findings such as trisomy 21, trisomy 8, monosomy 7, t(8;21), and complex karyotype."
Page 11, line 371: "Third, our study had a limited number of cytogenetic studies and lack of molecular and minimal residual disease (MRD) studies."
4. In the simple summary, there is the suggestion that current treatment protocols have to be modified. A discussion on suggested modifications, and also the specific approaches to managing hyperleukocytosis in this setting, would be necessary.
Response: Thank you for your comments. We have revised the manuscript accordingly.
Page 11, line 344: "Our opinion is that treatment recommendations for hyperleukocytosis in both AML and ALL patients should be modified to be more intensive in those who have identified risk factors to reduce complications secondary to leukostasis while being aware of the risk of developing consequences resulting from treatment by close monitoring and providing intensive care."
5. In the association of hyperleukocytosis with mediastinal mass, do you think it is through the association with T-cell immunophenotype or you see same association in B-ALL.
Response: In a multivariate analysis of ALL cases where the outcome is hyperleukocytosis, adding immunophenotype to a model containing mediastinal mass results in the significance for mediastinal mass disappearing. Thus, it is confounded by immunophenotype.
5. The study in some ways reinforce the idea behind the NCI risk criteria. While your cohort though sizable is still small, what do you think is the reason why the 50k-100k WBC group in ALL did not sure worse OS compared to reference. The threshold for the NCI is 50k.
Response: After adding immunophenotype to the multivariate model, the results changed. Therefore, we have revised the manuscript as follows:
Table 3. Multivariate analysis results showing independent risk factors for overall survival in childhood acute lymphoblastic leukemia patients
Risk factor |
Crude HR |
Adjusted HR |
P value |
Age (years) |
|
|
|
1 – 9 |
Reference |
Reference |
|
< 1 |
3.51 (1.79-6.89) |
3.05 (1.57-5.96) |
0.001 |
≥ 10 |
1.80 (1.29-2.52) |
1.64 (1.15-2.32) |
0.006 |
Male sex |
1.11 (0.84-1.47) |
1.37 (1.05-1.79) |
0.021 |
Immunophenotype |
|
|
|
T-cell |
Reference |
Reference |
|
B-cell |
0.79 (0.55–1.15) |
1.37 (0.90-2.08) |
0.14 |
FAB classification |
0.56 (0.31–1.01) |
0.85 (0.46-1.56) |
0.6 |
Initial WBC count (×109/L) |
|
|
|
< 50 |
Reference |
Reference |
|
50 – < 100 |
1.47 (0.97-2.23) |
1.59 (1.07-2.36) |
0.022 |
≥ 100 – < 200 |
2.45 (1.60-4.61) |
2.04 (1.33-3.14) |
<0.001 |
≥ 200 |
3.04 (2.00-4.61) |
2.71 (1.74-4.21) |
<0.001 |
Early complication |
1.16 (0.79-1.71) |
0.88 (0.62-1.26) |
0.5 |
Page 1, line 15: "Hyperleukocytosis, extreme hyperleukocytosis (WBC >200×109/L), age less than 1 year, age greater than 10 years, and male sex were independently associated with overall survival in the ALL group while extreme hyperleukocytosis and age less than 1 year were independently associated with overall survival in the AML group."
Page 1, line 35: "Hyperleukocytosis (hazard ratio [HR]: 2.04), extreme hyperleukocytosis (HR: 2.71), age less than 1 year (HR: 3.05), age greater than 10 years (HR: 1.64), and male sex (HR: 1.37) were independently associated with poorer overall survival in childhood ALL patients."
Page 6, line 185: "In the multivariate Cox regression analysis, risk factors for overall survival among the ALL patients are shown in Table 3. The independent risk factors for poorer overall survival were age, sex and WBC counts at initial presentation. Age less than 1 year and age greater than 10 years had hazard ratios of 3.05 (95% CI: 1.57 - 5.96) and 1.64 (95% CI: 1.51 - 2.32), respectively, when compared to those aged between 1 and 9 years (p = 0.001 and p = 0.006, respectively). Males had a hazard ratio of 1.37 (95% CI: 1.05 - 1.79) when compared to females (p = 0.021). Hyperleukocytosis and extreme hyperleukocytosis had hazard ratios of 2.04 (95% CI: 1.33 - 3.14) and 2.71 (95% CI: 1.74 - 4.21) compared with those who had an initial WBC count less than 50×109/L (p = 0.001 and p <0.001, respectively). Patients who had an initial WBC count of 50 to 100×109/L had a hazard ratio of 1.59 (95% CI: 1.07 - 2.36) compared with those who had an initial WBC count less than 50×109/L (p = 0.022)."
Page 10, line 328: "Our study found that on multivariate analysis, age, male sex, and WBC count at initial presentation were independently associated with overall survival in childhood ALL patients, consistent with results from other studies."
Page 11, line 375: "Hyperleukocytosis, extreme hyperleukocytosis, age less than 1 year, age greater than 10 years, and male sex were independently associated with a poorer overall survival in childhood ALL patients."
6. In the table 2, I think what is meant is induction of remission, and not remission of induction.
Response: We have revised the manuscript accordingly.
Variables |
Total (N=207) |
Hyperleukocytosis (N=42) |
No hyperleukocytosis (N=165) |
P value |
Treatment outcomes, n (%) |
||||
Induction of remission |
109 (67.3) |
19 (59.3) |
90 (69.2) |
0.289 |
Relapse |
63 (30.4) |
12 (28.6) |
51 (30.9) |
0.915 |
Mortality |
160 (77.3) |
34 (81.0) |
126 (76.4) |
0.669 |

Reviewer 3 Report
This is a well written paper reporting Acute leukemia (AML 207 and 483 ALL) in a large cohort of 690 children diagnosed from january 1998 to december 2017 and treated in the principal tertiary referral center in southern thailand.
Patients analysed retrospectively have been treated according to succesive European than Thai Pediatric protocol. Their outcome is properly analysed according to clinical risk factor and shows that overall outcome are poorer than in developed countries.
In this cohort, as in the litterature, well described, high hyperleucocytosis is associated with early complications and poor survival outcomes.
Age less than 1 year and greater than 10 are independently associated with poor survival.
The discussion is very long and should be shorten
none
Author Response
Reviewer #3:
Comments to the Author
This is a well written paper reporting Acute leukemia (AML 207 and 483 ALL) in a large cohort of 690 children diagnosed from January 1998 to December 2017 and treated in the principal tertiary referral center in southern Thailand. Patients analysed retrospectively have been treated according to succesive European than Thai Pediatric protocol. Their outcome is properly analysed according to clinical risk factor and shows that overall outcome are poorer than in developed countries. In this cohort, as in the litterature, well described, high hyperleucocytosis is associated with early complications and poor survival outcomes. Age less than 1 year and greater than 10 are independently associated with poor survival.
The discussion is very long and should be shorten.
Response: Thank you for your comments. Due to the journal's requirement that articles have a primary text of at least 3000 words and the fact that our cohort included patients with both ALL and AML, which should be discussed separately due to the heterogeneity of diseases, we made an effort to be concise and comprehensive.

Round 2
Reviewer 3 Report
the revised form is only a highlighed form on special point (for exemple cytogenetic has been done = my mistake , it is however strange that it does not have a pronostic impact